# Impact of Protein Intake on Training Response in Chronic Lung Disease

**DOI:** 10.3390/nu18010041

**Published:** 2025-12-22

**Authors:** Andrea Huhn, Patrick Diel

**Affiliations:** Department of Molecular and Cellular Sports Medicine, Institute for Cardiovascular Research and Sports Medicine, German Sports University, 50333 Cologne, Germany; diel@dshs-koeln.de

**Keywords:** chronic lung diseases, COPD, physiotherapy, strength training, nutrition, protein

## Abstract

Background/Objectives: Loss of muscle strength and mass is common in patients with chronic lung disease (CLD) and contributes to functional decline. Resistance training and adequate protein intake improve muscle function in healthy adults, but data for this patient population are limited. Methods: In this prospective, non-randomized controlled study, 16 patients with CLD (51–85 years) participated in three six-week intervention phases: phase 1—usual diet, phase 2—daily protein intake according to recommendations (1.2–1.5 g/kg bodyweight), and phase 3—additional protein supplementation immediately after training. Combined strength and endurance training was performed throughout the entire intervention period. The main outcome was maximum strength and body fat, while secondary outcomes included physical capacity, weight, activity, and quality of life (QoL). Data were analyzed using linear mixed-effect models to evaluate interaction effects between time points and phases with an intention-to-treat analysis. Results: In the patients’ usual diets, daily protein intake was below the recommended levels; during the intervention, protein intake increased significantly but did not reach the recommended target range. Maximum strength was increased marginally significantly by 4.6 kg during the intervention time without an interaction effect. Body fat reduction was significantly modeled using the interaction effect, whereas body weight remained unchanged. These enhancements are remarkable given that the training intensity was very low (less than once weekly), and protein consumption was below recommended levels. Conclusions: Targeted resistance training, combined with increased protein intake, led to measurable improvements in strength and body composition. These findings demonstrate that low-effort interventions can be implemented in real-life settings, providing a practical strategy to improve strength (a significant prognostic indicator) and increase protein consumption among this vulnerable population.

## 1. Introduction

Loss of strength and skeletal muscle mass is strongly associated with older adults and chronic diseases, including respiratory diseases [1]. Sarcopenia affects 26.8% (95% CI: 17.8–38.1) of this patient group, a prevalence double that of controls [2]. In chronic obstructive pulmonary disease (COPD), the most common lung disease, muscle wasting occurs in about 40% of patients and is potentially a sign of malnutrition [1]. This symptom is a key determinant of progression and pathophysiology in the circulus vitiosus of COPD [1]. Additionally, many normal-weight or even overweight patients are affected by sarcopenia [3] or low muscle mass [4].

Resistance training is known to be effective in increasing fat-free mass—even more so when combined with higher protein intake in healthy adults [4] and those with muscle wasting disorders [1]. Protein is a necessary component for the development of muscle cells and plays an important role in muscle protein synthesis [5]. Additionally, acute intake of protein immediately after training can optimize muscle protein synthesis after training and offer beneficial effects in recovery [6,7], including in COPD [8]. This factor could be relevant to patients with CLD since they often suffer from low training tolerance. Muscle protein synthesis increases up to 4 h in a fasted state and up to 24 h in a nourished state with high amino acids after training. In particular, 20–25 g of protein is needed to maximally increase muscle protein synthesis in young healthy adults [9]. This protein synthesis is further stimulated when protein ingestion is combined with carbohydrates and their resulting insulin secretion [10].

However, in COPD patients, systemic inflammation, increased protein catabolism, and anabolic resistance may blunt the anabolic response, suggesting that higher protein intakes of 25–30 g per meal could be necessary [1,4,9,11,12]. Referring to the ERS statement for nutritional assessment and therapy in COPD, “nutritional intervention is probably effective in undernourished patients and probably most when combined with an exercise training” [3]. It is known that body weight and body composition contribute to patient outcomes, independently from lung impairment [3]. It was also suggested to use nutritional supplements in addition to a balanced diet and exercise training in earlier stages of COPD due to its possible positive effects [3]. The ESPEN guidelines for pulmonary diseases state that “enteral nutrition in combination with exercise and anabolic pharmacotherapy has the potential to improve nutritional status and function“ based on four studies showing weight gain in nutritional interventions combined with pulmonary rehabilitation [13]. The PROT-AGE consensus statement recommends a protein intake of 1.0–1.2 g/kg bodyweight in healthy older adults (>65 years) and 1.2–2 g/kg bodyweight for chronic or inflammatory diseases [12]. Similar recommendations were made in the German guidelines for clinical nutrition (1.2–1.5 g/kg bodyweight in combination with physical activity) [14]. The optimal impacts on muscle strength occur at an amount of 1.5–1.6 g/kg bodyweight in adults, with no beneficial effects observed above this amount [5,15].

Effects were 1.7 times greater in those aged < 60 but related only to isotonic strength [5]. This factor is specific to training interventions and offers high muscle mass mobilization [5]. Therefore, previous studies using isometric strength tests as an indicator [16,17] may not have been sufficiently sensitive to detect relevant changes [18], highlighting the need for further research.

In a systematic review investigating add-on interventions in COPD, none of the studies using nutritional interventions investigated strength but only general physical capacity [19]. A review by Aldhahir et al. also showed that significant changes to nutritional interventions did not change physical capacity but led to significant changes in the same studies when investigating strength [17]. Strength, therefore, might be a more sensitive assessment point in such interventions. Nevertheless, changes were observed in only 3 of 11 studies with different nutritional interventions [17].

Previous studies were carried out in pulmonary rehabilitation or with novices in exercise training [16,17,19]. Since resistance training alone is a potent stimulus, combined interventions may mask the additive effects of protein supplementation among such populations [19]. In an exercise regimen, first, the training stimulus must be optimized; secondly, the daily protein intake must be adjusted; and third, the timing of protein intake must be considered [20].

Patients with COPD are more prone to increased visceral, but not subcutaneous, fat, which contributes to higher systemic inflammation [3]. Thus, body composition, not only body weight, should be evaluated. Moreover, the effect of protein intake on physical activity remains unclear. While about 45% of the German adult population meets the minimum recommendation of 150 min of moderate aerobic activity per week, this proportion is lower among individuals with COPD (approximately 33%) [21]. This value represents the lowest level of physical activity among those with non-communicable diseases, apart from those with depression [16]. COPD is thus a major factor contributing to reduced physical activity and limited engagement in muscle-strengthening exercises [21].

The effects of protein supplementation on muscle strength and physical function in patients with chronic diseases are still ambiguous [1,16,17,19]. Further studies investigating the additive effects of protein supplementation are needed to optimize patient management [19]. To address this knowledge gap, the present study will assess the impact of protein supplementation combined with resistance training on muscle strength and physical function in patients with chronic disease. By focusing on trained individuals in their usual environments using sensitive assessments, this study aims to clarify current uncertainties and provide evidence to inform clinical practice.

It was hypothesized that the training effects would be enhanced by increasing daily dietary protein intake and by providing additional protein immediately after each training session. Improvements were expected in the maximal strength of both the upper and lower extremities, body composition, physical capacity, physical activity, and QoL.

## 2. Materials and Methods

### 2.1. Study Design and Participants

A prospective, non-randomized controlled within-subject design was used. The study was pre-registered at the German Register for Clinical Studies (DRKS00029034) and ethically approved by the German Sports University Cologne. Patients were recruited from the physiotherapy program and specialized in lung diseases based on personal testimony and informational material. A minimum of 10 subjects was conservatively estimated with G*Power analysis (G*Power 3.1.9.7, Kiel, Germany) for maximum strength in the leg extensor (F-test Anova, 4 repeated measures within factors, a priori, effect size: f = 0.5; level of significance: Alpha < 0.05). In total, 19 patients provided written informed consent prior to their participation. With these assumptions, the actual power of the test was 0.80. Since the main outcomes were investigated seven times, the power increased to 0.98 for these measurements.

Inclusion criteria were diagnosed CLD except for Cystic Fibrosis and Primary Ciliary Dyskinesia and a valid prescription of ambulant physical training (“gerätegestützte Krankengymnastik”). Patients needed at least two weeks of training experience to guarantee familiarization [22]. Exclusion criteria were insulin-dependent Diabetes Mellitus, with acute exacerbation during the last month; acute infections; risk of fracture; restricted protein intake, intolerance to using food, functional food, or supplements to increase training effects; and severe psychological comorbidities or cognitive impairments that inhibited reliable food protocols unless a caregiver was available to administer help.

Sixteen participants were investigated, and 12 finished phase three. Two participants dropped out due to an end-of-training prescription. One participant was excluded from phase three because of his diabetes.

### 2.2. Study Intervention

The intervention included three different conditions of protein intake, each lasting six weeks. During all phases, patients tracked their daily nutrition and had prescribed physiotherapeutic training covered by German health insurance [23]. Training consisted of 60 min combined strength and endurance exercises with devices, which was usually prescribed 1–2 times per week. The endurance portion involved 10–15 min on an ergometry machine or treadmill, with the intensity set according to previous assessments. The strength portion included eight devices for all large muscle groups (see Appendix B), with 60 s of strain and 60 s break. This regimen was completed two times at intensity until momentaneous muscle failure. Training plans were individualized according to patient needs and progression and may have differed from the standard.

In phase one, participants practiced their usual training together with an individual conventional diet. The individual daily baseline protein intake was calculated. This phase was used as the control condition for each subject. In phase two, participants were advised to increase their daily protein intake to recommendations of 1.2–1.5 g/kg bodyweight. If patients already fulfilled the requested protein intake, the second condition was skipped. In phase three, patients continued receiving protein intake of 1.2–1.5 g/kg bodyweight. Additionally, after each training intervention, the patients received a shake containing 35 g whey protein (GermanProt 9000—Whey Protein Isolate, Sachsenmilch Leppersdorf GmbH, Wachau, Germany) and 32 g carbohydrates (Dextropur, Dextro Energy, Krefeld, Germany). This shake was consumed dissolved in 400 mL water within 30 min after finishing training under observation. Subjects were investigated at the beginning and end of each condition. Usually, the end of one condition marked the beginning of the next. An interruption in the study was possible after each phase if longer breaks (e.g., holidays) were expected. An additional measurement with only the main outcomes was completed after three weeks of each phase. One measurement of the interventions was conducted in advance to guarantee familiarization with the assessments and reduce learning effects. The study design is shown in Figure 1.

### 2.3. Outcome Measurements

The main outcome of the study was changes in maximum strength and body composition. To comprehensively assess the impact of protein supplementation on patients with lung disease, additional valid measures were recorded. These measures included physical capacity, measured with a six-minute-walk test (6 MWT) [24], handgrip strength [25], vital parameters, QoL questionnaires [26,27], and daily activity (daily steps and PASE questionnaires [28]). All tests are shown in Table 1 in the order of the testing procedure.

To determine the maximum strength, we used the indirect method of a one-repetition maximum (1 RM). This test is very reliable with a high ICC, even in clinical populations, and remains independent of age and exercise selection [29]. Participants engaged in a warm up of ten repetitions with their training weight. They were instructed to exhale during concentric movement to avoid the Valsalva maneuver. Weight was continuously increased until a maximum lower than eight repetitions was reached, with verbal encouragement. After each attempt, a 2–3 min rest was given [30]. The individual 1 RM was then calculated using the prediction equation of Kemmler et al. [31]. Body composition was analyzed with a four-point measurement body fat scale (Omron BF 511) at the same time during the day. Patients were asked to avoid consuming large meals or quantities of liquid at least 2 h before testing, and liquid intake was protocoled. The testing devices and (if applicable) standardized protocols are summarized in Appendix C.

During each phase of the intervention, patients recorded their daily diets via the app (Food Database GmbH, Bremen, Germany) or protocol (Freiburger Ernährungsprotokoll, Nutri-Science GmbH, Freiburg, Germany) according to patients’ preferences in the first and last week. The use of two assessment methods was permitted to ensure compliance in an elderly and clinically burdened population, where digital literacy varies considerably. Prior to study initiation, the comparability of both methods was evaluated in a pilot test, demonstrating no relevant differences in reported daily protein intake. Correct use of the chosen recording protocols was attempted together with the investigator during the first period of testing. During each intervention phase, three unannounced 24 h dietary recalls (two on weekdays and one on the weekend) were conducted by a trained investigator either via telephone or in person. Each recall interview lasted approximately 10–15 min and followed a structured, standardized recall protocol (see Appendix A).

At the beginning of phase two of the intervention, a short explanation on how to increase daily protein intake was given based on data from the patients’ nutritional records collected in phase one of the intervention. An information sheet with individualized daily protein intake and general instructions was given to the patients. A two-sided table providing information on how much of certain foods must be consumed to reach 10 g of protein was explained and given to the patients (Eiweissaustauschtabelle, Technical University Munich). The information sheet and table can be found in Appendix A. Individualized instructions on how to increase protein intake lasted about 10 min per participant. Patients were informed within 24 h if they did not reach their goal during phase two and three and were encouraged to further increase their protein intake. Advice regarding high protein food in the patients’ protocols was given as support.

During weeks one and six of each intervention phase, patients had to wear a step counter with a covered display for all seven days (Omron Walking Style IV) to quantify individual average physical activity. Participants were also allowed to use their own wearable devices and could occasionally assess their daily steps using them.

### 2.4. Data Analysis

Data were analyzed using intention-to-treat with linear mixed effect models. Phase one was used as a reference for individual physical changes without nutritional interventions. We hypothesized that patients would increase their maximum strength from pre to post measurement in each phase through physical training and that this effect would increase with higher protein intake. Therefore, the interactions between time point (pre, mid, and post) and intervention were stipulated as fixed effects. In addition, sex, age, and baseline FEV_1_ were included as fixed covariates, as these values were expected to influence strength performance. To account for individual differences in baseline strength and repeated measures per participant, random intercepts were included for each subject.

Missing data were handled inherently with the linear mixed-effects models implemented in the *nlme* package (R version 4.2.3, Vienna, Austria). These models use (restricted) maximum likelihood estimation, allowing the inclusion of all available repeated measurements under the Missing At Random (MAR) assumption. Participants with missing outcome values at specific time points remained in the analysis; no imputation procedures were applied.

The effect sizes of all parameters were evaluated with Cohen’s d. Fulfilment of the models’ assumptions was tested via residual plots (homoskedasticity) and Q-Q plots (Gaussianity of the residuals). Data were analyzed with R (version 4.2.3) and the *nlme* library.

## 3. Results

### 3.1. Anthromopometric Data and Protein Intake

An overview of participants’ anthropometric data is provided in Table 2. Detailed information regarding diseases and comorbidities can be found in Appendix D. The mean training experience in this setting was 5 years (3 weeks to 10 years).

Daily protein intake among the subjects is presented in Figure 2 and was under the recommended amounts (0.93 g/kg bodyweight, SD 0.24). Only one participant in phase one had a daily protein intake according to recommendations. Protein intake was highly significantly increased in phase two (*p* = 0.007) and phase three (*p* < 0.001), with large effect sizes (d = 1.12 and 1.52, respectively). Protein intake was increased by 0.13 g/kg bodyweight from phase one to phase two and by 0.05 g/kg bodyweight from phase two to phase three. Only 5 of 14 subjects reached their protein intake goals in phase two, while 5 of 12 reached their protein intake goals in phase three.

### 3.2. Strength

Results for the leg extensor are shown in Figure 3 and Table 3. Maximum strength increased marginally significantly from phase one to phase three (*p* < 0.1) and significantly from pre-measurement to mid-measurement (*p* < 0.05). Men had significantly higher strength values (*p* < 0.01) with high effects (d = 4.61). A higher age and lower FEV_1_ were significantly associated with lower strength (*p* < 0.05) with low effects. There was no significant interaction effect observed between time point and intervention. The raw mean increase in maximum strength from T1 to T4 was 4.6 ± 4.8 kg. The modelled strength changed by 4.0 kg, which represents an increase of 7.4%. At T1, three participants had a 1 RM < 120% of BMI, which is associated with higher mortality risk [32]. One participant increased their strength beyond this threshold during the intervention. The Mean Training frequency was 0.96 ± 0.29 times per week. Training intensity was 42.4 ± 10.2% of 1 RM at T1 for concentric weight.

For chest press, only 12 participants were analyzed due to the many shoulder problems in the investigated group. Additionally, only six female participants finished the measurements. The model yielded significantly high effects only for the factor of sex (*p* < 0.001). However, due to the missing data, no further analyses or graphical representations are presented for this variable.

### 3.3. Body Composition

Changes in body fat (Figure 4, Table 4) were modelled significantly and marginally significantly for the interactions of timepoint and intervention, respectively, in phase three with high effects, resulting in a 0.71% reduction in body fat. In contrast to body fat, changes in body weight were not significantly affected by the intervention or time points.

### 3.4. Secondary Outcomes

The 6 MWT, used as an indicator of physical capacity, showed a significant post-measurement effect with a moderate effect size (*p* < 0.05, d = 0.73). A significant and large effect was also observed in phase two (*p* < 0.05, d = 0.87). Age had a significant but small negative effect on physical capacity (*p* < 0.05, d = −0.29). FEV_1_ had a highly significant but small positive effect (*p* < 0.01, d = 0.14). The interaction between post-measurement and phase two was significantly negative with a large effect size (*p* < 0.05, d = −1.35). In hand dynamometry, both sex (*p* < 0.001) and age (*p* < 0.05) were significant predictors of grip strength. For physical activity measured with the PASE score, no significant factors were identified in the model. Changes in daily step count were not significant. No model parameters, including sex, age, FEV_1_, or phase, reached statistical significance. Changes in QoL were not significant, and no contributing factors reached statistical significance.

There were also no adverse events reported. Patients reported reversible sore muscles and fatigue as a result of the appointments. One participant reported diarrhea after drinking the first shake and was excluded from phase three, even though the causal relationship between the shake and illness was not clear.

## 4. Discussion

This study investigated the effects of modulating protein intake in combination with resistance training among individuals with CLD in their habitual environments. By focusing on trained individuals and applying sensitive isotonic strength assessments, our work expands previous research in the management of muscle dysfunction in CLD, which is primarily conducted in pulmonary rehabilitation settings with exercise novices [16,17,19].

Physical training is a highly effective strategy to improve prognosis. Reduced training tolerance due to dyspnea requires an optimized training stimulus in this target group to maximize effects. Since protein intake was expected to be lower than recommended and could inhibit training responses, the modulation of protein intake was used as add- on strategy in the current study. This study considered patients in their usual environments and provided information about the effects of modulating protein intake under real-life conditions. Unlike previous studies, our intervention followed a two-step strategy—first, by adapting participants’ habitual diets, and second, by introducing additional protein supplementation after training. Evidence from healthy adults indicates that the effect of increased protein intake on fat-free mass is significant among trained individuals but blunted in older adults [15]. Investigating trained patients allowed us to specifically evaluate the additive effects of nutritional support, in contrast to previous studies with combined interventions in untrained individuals.

### 4.1. Dietary Changes

In agreement with our hypothesis, subjects’ daily protein intake was below nutrition recommendations at the beginning of the intervention. Only one individual fulfilled the requirement of 1.2 g/kg per day. Our observed low protein intake in the daily nutrition of patients with lung disease also agrees with the literature. A previous study investigating 79 patients with COPD showed that 15% had a protein intake lower than 1.2 g/kg per day [33].

Although the information provided on the protein content of foods and personal diet reflection was very basic, it led to significant changes in eating behaviors. The observed increase in protein intake corresponded to an amount equivalent to two eggs (10 g) per day for the average participant and was maintained throughout the study. For comparison, a large meta-analysis in healthy individuals reported an average protein intake increase of 0.3 ± 0.5 g/kg per day but used supplementation, exceeding the increase achieved in our study [15]. While our increase did not reach the clinically relevant threshold of 0.25–0.3 g/kg body weight per day needed to stimulate muscle protein synthesis [34], it is noteworthy that an intervention requiring minimal effort was sufficient to produce measurable dietary changes.

### 4.2. Primary Outcome

The results showed a non-significant increase in maximal lower-limb strength. Without specific training, the decline in isometric quadricep strength is 2–4 times higher (4.3 vs. 1–2% per year) among those with COPD than within the healthy aged population [35]. Despite the low training frequency of once per week, the present intervention resulted in a considerable, though not statistically significant, improvement of four kilograms in 1 RM over 18 weeks, highlighting the effectiveness of targeted training. Considering the long training experience of this subject, this result is especially remarkable since most adaptations occurred at the beginning of a training intervention in this group. In comparison, a large meta-analysis showed an increase of 2.49 kg in 1 RM over 13 ± 8 weeks with resistance training and protein supplementation (36 ± 30 g per day) in healthy adults, with protein expected to augment the training effects by only approximately 9% [15]. The add-on effect of nutritional intervention, however, was not verified in our study. The difference between the raw and modelled increases in maximum strength was small (4.6 kg vs. 4.0 kg). This slight reduction in the modelled values reflects adjustments for individual differences, including age, sex, and baseline strength; accounts for missing data; and does not diminish the validity of the intervention effects.

### 4.3. Protein Supplementation

Protein supplementation in phase three was specifically optimized to the needs of the target population, aiming to optimize muscle protein synthesis [6] and account for the expected changes in mostly older patients with lung disease [11]. According to previous results [8], an increased training response was expected as the result of increased pro-regenerative effects caused by the protein supplementation immediately after training. Significant interaction effects were not verified. Due to the very low training frequency, and, therefore, low frequency of application, this result is not surprising. Moreover, in contrast to studies using supplements enriched with micronutrients [17], in our study, pure whey protein and glucose were used, ensuring that the observed effects could be attributed solely to these components.

### 4.4. Secondary Outcomes

The results from the 6 MWT showed significant improvements at post-measurement, with a moderate effect size. An even higher effect was observed in phase two, with negative interactions between timepoint and phase. Given the low frequency of testing in contrast to strength tests and missing data, these fluctuations could partially reflect daily variability in patient performance, rather than solely the effects of the intervention. These results support our hypothesis that physical capacity is less sensitive to resistance training than strength measures.

No significant effects were observed for overall physical activity measured with the physical activity score for the elderly (PASE) and daily step counts or QoL, suggesting that the intervention did not elicit measurable changes in broader functional or behavioral outcomes within the study period. Longer or more intense interventions might be necessary to achieve the relevant changes.

In the applied models, sex, age, and baseline FEV_1_ were included as fixed covariates. Sex had a large effect on outcomes in all strength tests. However, the effect of lower FEV_1_ was only marginally significant for the lower extremity strength and 6 MWT and remained undetectable in the other outcomes. These results are in line with previous findings [36] and indicate that although disease severity plays a role, physical performance is more strongly determined by training status.

### 4.5. Strengths

The present study combined strength-specific training, sensitive strength assessments, and nutritional intervention. The absence of significant improvements in non-strength performance measures highlights the specificity of the training effects, confirming that resistance exercises primarily enhance strength-related outcomes [37]. We suggest that previous studies may not have focused on outcomes clinically relevant to the intervention [19]. For instance, some investigations did not include any strength testing or relied on isometric strength assessments, which are less sensitive to training-induced changes [38].

It is known that nutritional interventions play an increasingly large role in optimized training stimuli [20]. Although the training frequency was low in our study, all patients practiced regular training during the complete intervention period and were already accustomed to training before the start of the intervention. This factor may offer advantages compared to previous studies, which were performed in pulmonary rehabilitation settings with training novices, often using multifactorial intervention [17]. Training is a highly effective method that can overlap nutritional effects when practiced for the first time [19].

In the present study, all subjects were investigated under three different conditions of protein intake to guarantee comparability and avoid group differences since patients differed strongly in their characteristics. High testing frequency reduced bias caused by the daily variability in the individual health status of the patients. The observed improvements in strength and dietary behavior were particularly remarkable given the challenges of working with older patients with lung disease, a population that is difficult to recruit and retain in intervention studies.

The study proved to be safe and practically feasible in this patient population. The implementation of increased protein intake did not pose any health risks, as the amount provided was consistent with current dietary recommendations for older adults and patients with lung disease. Patients with restricted protein intake were not included in the study. Even relatively low-effort interventions resulted in measurable improvements with high effects in daily protein intake, maximal strength, and body composition.

These findings demonstrate that targeted training and nutritional strategies can be effectively implemented under real-life conditions in this vulnerable group, even when training frequency and intensity are modest.

### 4.6. Limitations

Nevertheless, this study has several limitations. For example, confounding factors may have influenced the results due to the pragmatic study design. This study was conducted using a one-size-fits-all design whereas individualized approaches might yield more successful outcomes. More specific inclusion criteria might also be helpful to reduce inter-individuality [19]. A multicenter approach, moreover, could be a valuable alternative to increase external validity. Food diaries are known to change eating behaviors and might influence results [39]. Nevertheless, the significant reduction in fat mass but not weight can be interpreted as an indicator for the effectiveness of the intervention, not only as reporting bias. This result is especially remarkable because changes in body composition are known to occur slowly with resistance training and protein supplementation [15]. However, the modelled reduction is not considered to be clinically relevant.

Apart from amount, protein quality and timing of intake are known to affect protein synthesis [11]. In the present study, only the overall quantity of protein was assessed, without accounting for general dietary composition or evaluating potential nutrient deficiencies. However, different protein sources vary in their biological values. Factors such as digestibility, amino acid profile, and bioavailability are central determinants of protein quality and may lead to distinct effects on muscle protein synthesis and functional outcomes. These factors should be considered in future studies to better understand how protein interventions can be optimized for patients with chronic lung disease.

In addition, while whey protein primarily targets skeletal muscle, it may also modulate systemic inflammation in patients with COPD [40], an aspect that was not evaluated in this study.

## 5. Conclusions

This study demonstrates that even low-frequency resistance training—performed less than once per week with experienced subjects—can lead to measurable gains in lower-extremity strength for patients with CLD. Moreover, simple and low-effort nutritional education was sufficient to substantially increase daily protein intake, even though participants did not fully reach the targeted range of 1.2–1.5 g/kg bodyweight. Under these conditions, no significant additive effect of increased protein intake on training outcomes could be verified in the intention-to-treat analysis. Nevertheless, this approach remains worth pursuing, as such effects may emerge in populations that achieve the targeted protein range. Given the minimal educational requirements and low burden of the program, this practical strategy could be readily integrated into routine care and may offer meaningful benefits for patients and the healthcare system.

## Figures and Tables

**Figure 1 nutrients-18-00041-f001:**
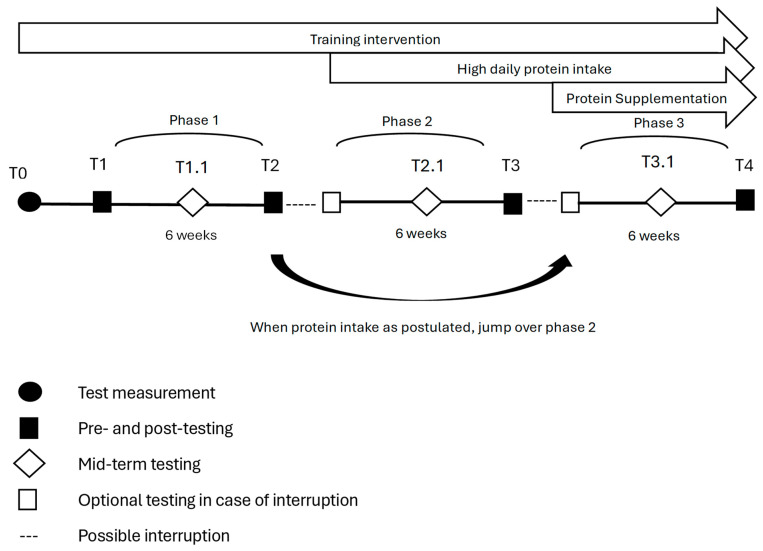
Interventions and testing in the prospective controlled study design.

**Figure 2 nutrients-18-00041-f002:**
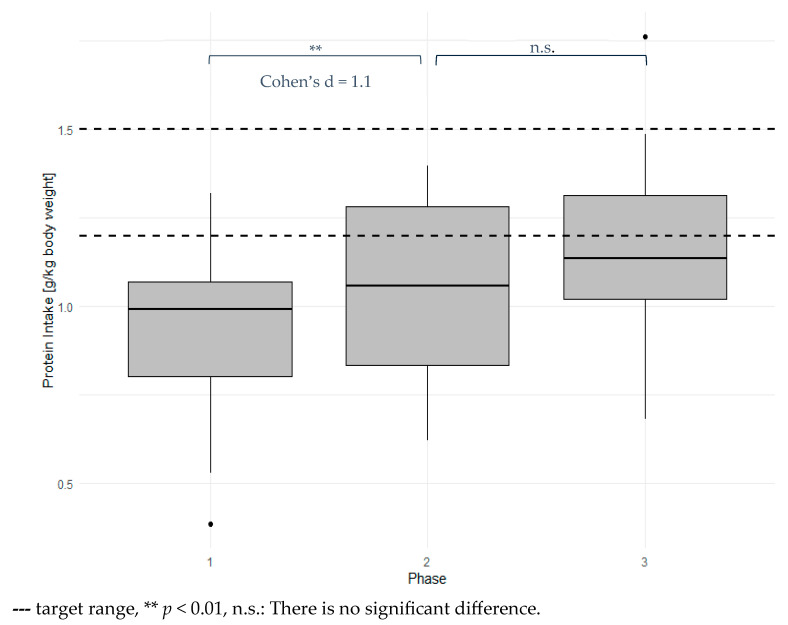
Mean daily protein intake for each phase.

**Figure 3 nutrients-18-00041-f003:**
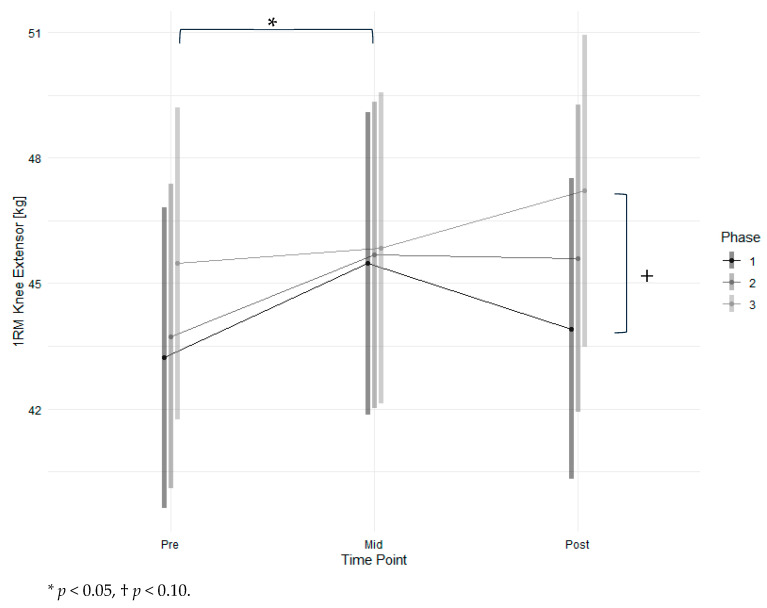
Estimated marginal means of 1 RM for the knee extensor across time points and study phases with 95% confidence intervals.

**Figure 4 nutrients-18-00041-f004:**
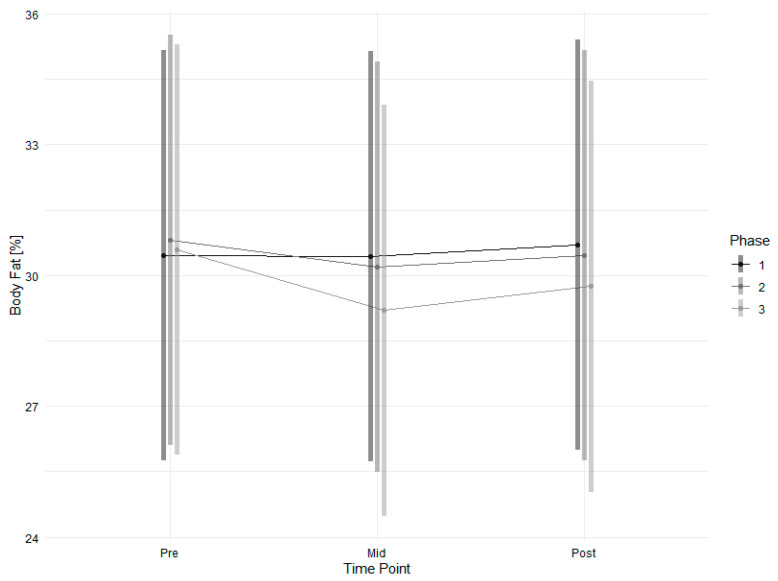
Estimated marginal means of body fat across time points and study phases with 95% confidence intervals.

**Table 1 nutrients-18-00041-t001:** Assessments at each testing point (in order).

	T0	T1	T1.1	T2	T2.1	T3	T3.1	T4
Vital values	X	X	X	X	X	X	X	X
Body weight, body composition	X	X	X	X	X	X	X	X
One-repetition maximum (leg extensor, chest press)	X	X	X	X	X	X	X	X
Hand dynamometry	X	X	X	X	X	X	X	X
Activity questionnaire (PASE)	X	X	X	X	X	X	X	X
Obstruction (FEV_1_)	X	X		X		X		X
Questionnaires QoL (Mmrc, SF-36, CAT)	X	X		X		X		X
Six-minute-walk test		X		X		X		X

FEV_1_—Forced Expiratory Volume in 1 Second; CAT—COPD Assessment Test.

**Table 2 nutrients-18-00041-t002:** Anthropometric data.

Variable	Mean (SD)	Min–Max
Age (years)	66 (8.1)	51–85
Sex (m:f)	5:11	
BMI (kg/m^2^)	26.3 (5.3)	19.7–38.1
FEV_1_ (%)	60 (22.0)	17–86
6 MWT (m)	464 (92.8)	291–610
Steps/day	4548 (3086)	766–10,062
Symptom burden (CAT score)	15 (7.5)	3–31

**Table 3 nutrients-18-00041-t003:** Linear mixed-effects model coefficients for knee extension 1 RM (kg) across phases and time points.

	Estimate	Std. Error	t-Value	*p*-Value	Cohen’s d
(Intercept)	53.95	13.28	4.06	0.002	
Mid	2.24	1.05	2.12	0.035 *	0.76
Post	0.69	1.04	0.66	0.51	0.23
Phase 2	0.51	1.08	0.47	0.640	0.17
Phase 3	2.25	1.71	1.92	0.058 †	0.76
Sex	13.56	3.23	4.20	0.001 **	4.61
Age	−0.45	0.19	−2.36	0.037 *	−0.15
Base FEV_1_	0.20	0.07	2.94	0.013 *	0.07
Mid/Phase 2	−0.30	1.55	−0.20	0.845	−0.10
Post/Phase 2	1.17	1.54	0.76	0.448	0.40
Mid/Phase 3	−1.88	1.65	−1.14	0.257	−0.64
Post/Phase 3	1.04	1.63	0.64	0.523	0.35

* *p* < 0.05, ** *p* < 0.01, † *p* < 0.10.

**Table 4 nutrients-18-00041-t004:** Linear mixed-effects model coefficients for body fat (%) across phases and time points.

	Estimate	Std. Error	t-Value	*p*-Value	Cohen’s d
(Intercept)	63.14	18.26	3.46	0.005	
Mid	−0.03	0.42	−0.06	0.952	−0.02
Post	0.24	0.42	0.57	0.569	0.20
Phase 2	0.36	0.43	0.82	0.414	0.30
Phase 3	0.13	0.48	0.28	0.778	0.11
Sex	−10.69	4.45	−2.40	0.033 *	−9.10
Age	−0.40	0.26	−1.50	0.159	−0.34
Base FEV_1_	−0.02	0.09	−0.24	0.811	−0.02
Mid/Phase 2	−0.58	0.61	−0.96	0.341	−0.50
Post/Phase 2	−0.58	0.61	−0.95	0.346	−0.50
Mid/Phase 3	−1.37	0.65	−2.10	0.039 *	−1.16
Post/Phase 3	−1.08	0.64	−1.68	0.097 †	−0.92

* *p* < 0.05, † *p* < 0.10.

## Data Availability

The raw data supporting the conclusions of this article will be made available by the authors on request.

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
