# Peer review of "Impact of Protein Intake on Training Response in Chronic Lung Disease"

_nutrients, 2025, doi:10.3390/nu18010041_

Round 1
Reviewer 1 Report
Comments and Suggestions for Authors
Overall evaluation:
This study was conducted to determine whether the positive effects on muscle strength and physical function of protein supplementation in patients with chronic disease. The research direction is in the field of clinical nutrition, which conforms to the scope of the journal, is close to practical application, and the results have certain clinical guiding significance. However, the picture quality of the paper is general, and the language logic and hierarchy need to be strengthened. In addition, there are still some problems to be improved.
Specific modification suggestions:
1.Line54, "25-30g per meal", this unit is not scientifically rigorous, it is suggested to change to "g/kg bodyweight".
2.Line74-78, this part of the content is far from the core content of this paper, it is suggested to delete or simplify.
3.Line92-95, research content and methods should be supplemented, and the description of expected research results should be deleted, which can illustrate the significance of carrying out this study.
4. The "Materials and Methods" and "Results" of the manuscript should be added with secondary headings to make the manuscript clear.
5.Table1, it is suggested to add "column name" to the left side of the first row of the table, and "Age,CAT" does not indicate the unit.
6. This study adopted an increase in protein intake to improve chronic lung disease. However, according to the existing research results, chronic obstructive pulmonary disease can lead to renal function decline, and then cause kidney disease. Excessive protein intake can aggravate nephropathy, which contradicts the findings of the present study and needs to be discussed in the manuscript.
7. This study only focused on the relationship between protein intake and training response in chronic lung disease, but in fact, different proteins have different biological effects due to their different digestibility and utilization rate. It is suggested to increase the discussion in this aspect.
Reviewer 2 Report
Comments and Suggestions for Authors
My recommendations are as follows:
Abstract: I recommend mentioning whether sessions targeting endurance and strength were performed in the patients during the 3x6 weeks. I recommend mentioning the range or average age of the patients. In the Methods section, I recommend mentioning which parameters were targeted for physical capacity. I also recommend mentioning the quality of life and activity testing instruments. I recommend mentioning which instruments were used to determine the parameters targeting muscle mass, body fat, etc. In the Results section, I recommend mentioning the significance level found, numerically. The results do not mention anything regarding secondary outcomes, I recommend clarifications. I recommend rewriting the purpose of the present study.
Keywords - mention COPD, I recommend writing descriptively, especially since it is not mentioned in the previous text.
Lines 72-73 mention - Previous studies, but only one bibliographic index is presented, I recommend clarifications. Idem 90-91
Lines 75-78 recommend mentioning the bibliographic sources you refer to, especially since it is a paragraph.
I recommend mentioning the purpose of the study, and the hypotheses to be reformulated.
2. Materials and Methods – I recommend reorganizing it into subsections, not as a block text. Subsections: Study design, Participants, Procedures, Evaluation instruments, Data analysis, etc.
Lines 139-140 repeat the idea, I recommend deleting it.
Lines 140-143 recommend detailing the instruments used. I recommend mentioning the reliability of the questionnaire. In appendix 2, they are only presented without referring to the standardization of the instruments, and to the scales for assessing the results. I recommend rewriting.
In conclusion, due to the way this section is organized and presented, it is difficult to understand, as I mentioned before, I recommend dividing the text into subsections and mentioning the complete information specific to a scientific article.
Lines 187-189 recommend moving to the Participants section. Lines 190-191 move to the Participants subsection.
Table 1 recommends mentioning the standard deviations for the mentioned indicators. I recommend that all the acronyms used be mentioned descriptively under table 1.
Table 2 this indication - , ***p < .001, does not appear, I recommend deleting it.
I recommend that the Discussion section be organized according to the rewritten hypotheses, on the targeted parameters. Also, some of the mentioned aspects targeted in the study such as quality of life etc. are not correlated nor presented in the Results section.
In conclusion, I recommend expanding this section by making new concrete correlations between the results of this study with results from previous studies and reorganizing it.
In all sections, there is no logical coherence in the presentation of the aspects concerned, there is a rush through text and information.
I recommend a major revision to the authors before resubmitting.
Reviewer 3 Report
Comments and Suggestions for Authors
Dear Authors,
Thank you for providing me with the opportunity to read this manuscript. Below, I have listed my comments:
1) Introduction- Several grammatical errors (is strongly associated to older adults, muscle waste occurs about 40%, Circulus virtuosos, an protein intake). It is best to proofread the paper.
2) the theoretical framework for your intervention could be clarified more.
3) The introduction needs a stronger connection to the research gap. Highlight what is missing in the literature and why your study fills this gap.
4) Lines 92–94- Good hypothesis, but phrasing unclear (modulated be increased protein intake). The outcomes are also listed but not organized. You could group outcomes (strength, body composition, function, quality of life).
5) The Materials and Methods section would benefit from subheadings that could guide the reader.
6) The design is described as prospective, non-randomized controlled, but the control condition is not clearly defined. a) Add a clear definition of the control condition. b) Explicitly state whether subjects act as their own controls (within-subject design). c) Clarify if the order of phases was fixed or counterbalanced (it appears fixed).
7) The explanation of power analysis is vague: What outcome variable was used?What effect size (0.5) corresponds to? Actual power is not reported.
8) In the outcome measures, you could specify units for outcomes (e.g., kilograms for strength, kg/m² or % for body fat).
9) Dietary assessment- You used two different methods (app vs protocol) but only briefly mention comparability. Mid-phase recalls are mentioned but not standardized. It is best to provide details about Who performed recalls? How long were recall interviews? Were recalls validated? How were missing data handled? Further, provide reasoning for allowing two different tracking methods.
10) Physical activity measurement- could you please clarify the following? Was the accelerometer worn for 7 days each time or only for the day of measurement?Were step counts averaged? How were non-wear periods defined?
11) In the analysis, how were missing data handled (e.g., maximum likelihood?)
12) In the discussion, the argument that 'even low-effort interventions produced measurable effects' is interesting but not well supported by quantitative data.
You could insert specific effect sizes or means to strengthen this point.
13) Literature is referenced but not always clearly connected to your data.
Compare your findings more explicitly to prior studies (e.g., In contrast to Smith et al., who found X, our results indicate Y).
I hope thsi feedbakc is helpful.
Round 2
Reviewer 2 Report
Comments and Suggestions for Authors
No comments
Reviewer 3 Report
Comments and Suggestions for Authors
Thank you Authors for revising the manuscript.